# Effectiveness of lumbar support with built-in massager system on spinal angle profiles among high-powered traffic police motorcycle riders: A randomised controlled trial

Nur Athirah Diyana Mohammad Yusof[1], Karmegam Karupiah[1]*, Shamsul Bahri Mohd Tamrin[1‡], Irniza Rasdi[1], Vivien How[1‡], Sivasankar Sambasivam[1‡], Putri Anis Syahira Mohamad Jamil[1], Kulanthayan K. C. Mani[2‡], Hassan Sadeghi Naeini[3‡], Dayana Hazwani Mohd Suadi Nata[4‡]

1 Faculty of Medicine and Health Sciences, Department of Environmental and Occupational Health, Universiti Putra Malaysia, UPM Serdang, Selangor, Malaysia, 2 Faculty of Medicine and Health Sciences, Department of Community Health, Universiti Putra Malaysia, UPM Serdang, Selangor, Malaysia, 3 Industrial Design Department, School of Architecture & Environmental Design, Iran University of Science & Technology, Tehran, Iran, 4 Faculty of Health Sciences, Center for Toxicology and Health Risk Studies, Universiti Kebangsaan Malaysia, Kuala Lumpur, Malaysia

☯ These authors contributed equally to this work.
‡ These authors also contributed equally to this work.
* megam@upm.edu.my

## Abstract

Traffic police riders are exposed to prolonged static postures causing significant angular deviation of the musculoskeletal, including the lumbar angle (L1-L5). This postural alteration contributes to awkward posture, musculoskeletal disorders and spinal injury, especially in the lower back area, as it is one of the most severe modern diseases nowadays. Thus, the study aimed to evaluate the effect of lumbar support with a built-in massager system on spinal angle profiles among traffic police riders. A randomised controlled trial (pre-testpost-test control design) was used to assess spinal angle pattern while riding the high-powered motorcycle for 20 minutes. Twenty-four traffic police riders were randomly selected to participate and 12 riders were assigned to the control group and 12 riders to the experimental group. The pre-test and post-test were conducted at a one-week interval. Each participant was required to wear a TruPosture Smart Shirt (to monitor spinal posture). The TruPosture Apps recorded the spinal angle pattern. The data indicated that the police riders using motorcycle seat with lumbar support and built-in massager system showed a huge improvement in maintaining posture which only involves slight spinal angle deviation changes from the spinal reference angle throughout the 20 minutes ride. The data collected then were analysed using the Mann-Whitney test and Wilcoxon signed-ranked test to verify a statistically significant difference between and within the control and experimental groups. There were significant differences in all sensors between the control group and experimental groups (p<0.05) and within the experimental group. According to the findings, it can be said that the ergonomic intervention prototype (lumbar support with built-in massager system) successfully helps to maintain and improve the natural curve of the spinal posture. This

**Data Availability Statement:** All relevant data are within the manuscript and its Supporting information files.

**Funding:** This research is supported by the Ministry of Education Malaysia (MOE) through Fundamental Research Grant Scheme (FRGS/1/2015/SKK06/UPM/02/2), Graduate Research Fellowship (GRF), Universiti Putra Malaysia, IPS Putra Grant (9616000), and S-GRA (6300260). No funding bodies had any role in decision to publish or preparation of the manuscript.

**Competing interests:** The authors have declared that no competing interests exist.

indirectly would reduce the risk of developing musculoskeletal disorders and spinal injury among traffic police riders.

## Introduction

The riding posture is very important because an awkward riding posture may cause discomfort to the riders and increase the risk of developing musculoskeletal disorders (MSD) in long-term exposure. De Cássia et al. [1] stated that body discomfort is an indicator of many MSD due to prolonged discomfort. A sustained flexion of lumbar lordosis (low back) during driving usually is the main factor in the generation of low back pain [2]. According to Makhsous et al. [3], sitting decreases lumbar lordosis compared to the standing posture, resulting in increased disc pressure and low back muscle activity. This is because while sitting, the ischial tuberosity mainly supports the majority of the upper body weight. Increased pressure in this area is significantly associated with increased spinal load [4]. Riding a motorcycle exposes the officers to excessive physical demand, especially for prolonged periods. They will tend to feel discomfort and possibly fatigue while sitting in the same position with restricted movements. Thus, to improve spinal posture and comfort in transportation settings, seat design has a significant role and efficiency.

A well-designed seat would decrease pressure within the spinal discs, spinal ligaments, and gluteal muscle [5–8]. Lumbar support is the solution for the problem, whereby the weight and pressure of the trunk are taken by the back support and increased lumbar lordosis. The use of a vehicle seat (lumbar support) has been significantly associated with the reduction of muscle discomfort and low back pain during prolonged riding and driving journey [5,8]. Adjustable lumbar support with extra cushioning, which provided a massage-like comfort, has been proved to reduce transmission of vibration, which indirectly increases muscle comfort by improving oxygen and blood flow to the tissue [9,10]. According to Mansfield et al. [11], the seat shape, user suitability, seat material, duration of sitting in the same position, vibration, and posture changes are among the factors that influence seat comfort.

Traffic police is a police officer who serves to enforce the rules related to traffic. They have to perform various work tasks, such as escorting important person to any events, patrol selected locations, and finding any road offenders. All this work task requires them to use a motorcycle as their vehicle. At present, the officer riding a motorcycle takes at least five hours per day during duties. Therefore, traffic police riders have to endure prolonged riding duration. Rashid et al. [12] believed that continuous riding of a standard motorcycle for an extended time would result in a high level of postural fatigue and health problems. From a previous research study on the CBX 750P21 motorcycle, it was found that 88.3% of traffic police riders suffered from MSD with 34.3% of them suffering from lower back pain due to static posture and prolonged sitting while riding the motorcycle [13]. Another previous study also revealed that more than half (54.7%) of the traffic police riders rode high-powered motorcycles for an average of 5.64 hours per day with a fixed posture leading to increased discomfort from prolonged sitting that enhanced muscle fatigue [14]. This showed that most of their working time involves riding motorcycles. However, to the best of the authors' knowledge, no research has been done on the spinal riding posture during a motorcycle ride, especially among police officers in Malaysia who use a motorcycle to perform their duties daily.

To solve this problem, ergonomics is the best solution for scientific research in man-machine interaction at the workplace since this field involves fitting machine to workers comfortability to improve their working performance, reduce fatigue, and stress [15]. The ergonomic application is very significant in areas involving prolonged and static riding activities

that directly affect the riders' healthy spinal posture and reduce muscle fatigue. Lumbar support with a built-in massager system is an intervention in this study to solve the ergonomic issue among motorcycle riders. The use of a back (lumbar) support has been proved to help maintain the natural spine lordotic lumbar curvature of the person while sitting compared to those without back support [16]. In a clinical and laboratory setting, automobile seat massage is utilised widely, and it has been proved efficacious in recovering from postural fatigue [9,10]. However, research and data on the effectiveness of this intervention seat, lumbar support with a built-in massager system, in an in-field setting (on-the-road) are insufficient. Thus, the present study integrates a massager system and lumbar support with the existing seat which an on-road test was conducted to maintain and support the spinal posture of the riders more efficiently as the test involved prolonged and static riding. In this study, the spinal angle profile was measured and assessed to determine the effectiveness of theprototype seat in supporting the spinal body posture of the riders throughout 20 minutes motorcycle ride. Thus, the present study seeks to evaluate the effect of lumbar support with a built-in massager system (prototype seat) among traffic police riders.

## Materials and methods

### Study design

A randomised controlled trial, pretest-posttest control group design was conducted among 24 traffic police riders who ride a high-powered motorcycle (Honda CBX 750). A simple random sampling was used in this study which the subjects were randomly assigned to the control group (12 riders) and experimental group (12 groups). Data collection commenced in March 2020 and finished in July 2020.

### Eligibility criteria

Eligible subjects were male riders only because almost 90% of traffic police riders are male. BMI was between 18.5 to 29.9 kg/m$^2$. The age group recruitment was between 20 and 39 years old because more than 40 years old are usually prone to experiencing low back pain due to the age factor and changes in spinal posture [17]. At least one year of experience in high-powered motorcycle riding. Exclusion criteria were the presence of any injury (under treatment or taking any medication related to muscle pain), especially lower back pain, in the 12 months preceding this survey.

### Sample size

To calculate the appropriate sample size, we use the formula of group comparison guidelines in a study done by Donnelly et al. [9] as a reference. The estimated SD (30.2), estimated larger mean (57.2) and estimated lower mean (19.0) were used in this formula. In this study, the desired power is 80% and the significant level is 95%. Thus, each experimental and control group would have 10 subjects. However, an additional 20% dropout rate was added in the sample size calculation. Thus, each experimental and control group will have 12 subjects. Therefore, a total of 24 subjects was selected in this study.

### Participant recruitment

The recruitment strategy involved taking the name list of all officers working at the Kuala Lumpur Traffic Police Station. The name lists were obtained from the Human Resources Department. However, 97 officers from the escort department only were chosen to ensure the consistency of data in which different department have different task or job and the duration

of riding police's motorcycle. Only 24 respondent who fulfilled the criteria were selected for the study. For qualification criteria, respondents were asked to fill up the pre-survey form and the BMI were measured by the main researcher. After the eligible subjects were selected, then the main researcher randomly assigned 24 of them into control and experimental group. This study's sampling method was simple random sampling, in which all the subjects' names were numbered into pieces of paper and put in a container. Then, the subjects were chosen randomly into the experimental or control group by Fishbowl Technique. Each subject in the experimental group received lumbar support with a built-in massager system during post-test. The pre-test and post-test were conducted at a one-week interval. Detail about study flow was illustrated in the CONSORT diagram, Fig 1.

Subjects (n = 12) allocated into the control group had undergone post-test session without the lumbar support with a built-in massager system (existing seat) during 20 minutes riding session. On the other hand, the experimental group (n = 12) had undergone a post-test session with the lumbar support with a built-in massager system (prototype seat) for 20 minutes riding session.

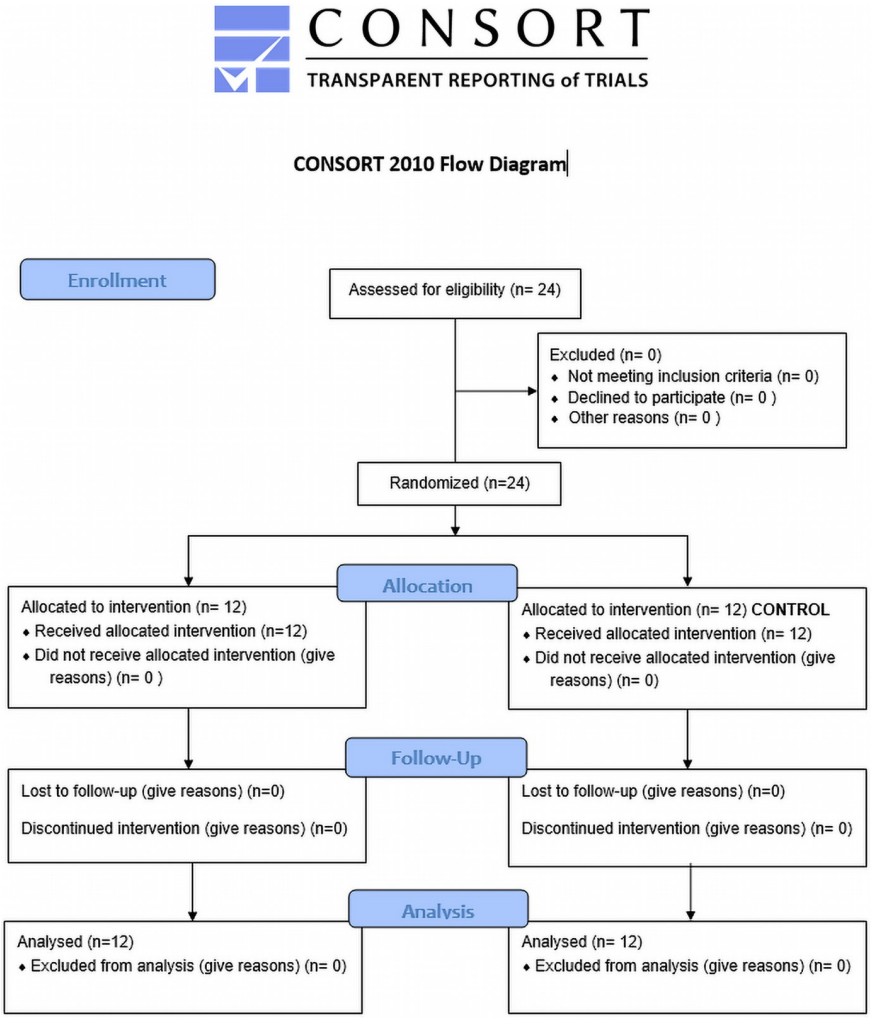

**Fig 1. CONSORT diagram.**

Before the experimental session began, subjects were allowed to choose the Truposture smart shirt that fits their body. This was done to ensure the point of each sensor would be placed in the correct position of the spine. There were no cases of unable to follow up or incomplete experimental testing as planned in this study. All subjects completed the follow-up stage (n = 24) in both intervention groups. This study had no excluded analysis cases as all the data were successfully recorded without any missing data.

## A prototype

The lumbar support on the prototype seat can be adjusted either upward and downward according to the lumbar height position for the comfortability of the riders [8]. There is also a massage fitted inside the lumbar support with two rotating balls [10]. The lumbar massage was set at a ratio of 1 minute on and 4 minutes off [18]. It was used for 20 minutes in the experimental riding session [19]. It can support up to 150 kg body weight.

## TruPosture

A spinal posture angle was obtained using the TruPosture Smart Shirt connected with the TruPosture mobile app among the police riders while riding a motorcycle with an existing seat for both groups; control and experimental during the pre-test. Then, during the post-test, the control group used the same motorcycle seat, and the experimental group used a prototype seat containing adjustable lumbar support with a build-in massager system. The TruPosture Smart Shirt covered different regions of the spinal alignment, with five nano-sensor technology attached undershirt. The five sensors cover the thoracic vertebrae, lumbar, and the pelvic regions (T1, T8, L1, L3, and Pelvis).

The technology helped the researcher monitor the entire spinal's alignment and curvature in several modes, including standing and sitting. The TruPosture mobile apps interface (Fig 2) and Windows software were utilised as these tools help the researcher track the posture and record the movement of posture in real-time. The sensors detect the real-time changes of spinal posture angle based on the spinal movement.

Based on Fig 2, the blue curve represents the spinal reference posture (ideal posture) at 0 minutes, and the orange curve indicates the actual spinal riding posture throughout 20 minutes. The number on the left and right side of the interface represents the angle reading for each spinal point sensor. The positive and negative values of angle indicate the spinal position, which is forward and backward respectively.

The validity of the equipment had been tested and approved by the previous study which was suitable for monitoring the posture of the spine [20]. The reason for choosing this method is that the spinal sitting pattern studies were limited to laboratory and clinical settings only [2,21]. According to Ma'arof et al. [22], the research field provides vital outcomes and real-world motorcycling information. Thus, TruPosture Smart Shirt is the most suitable method to monitor the spinal change pattern among motorcyclists on-road with the application of real working conditions.

## Data collection

In this study, the spinal reference posture at the 0 minutes riding session was set first. The subjects wore the TruPosture Smart Shirt during the monitoring, according to the ideal riding posture. An upright riding posture is the best posture in riding this type of motorcycle based on the standard operating procedure (SOP) of Honda CBX750P motorcycle riding. According to Ma'arof et al. [22], a good riding posture for the upright posture is that the arms should bend slightly when gripping the throttle and sitting in the upright position. The shoulders and

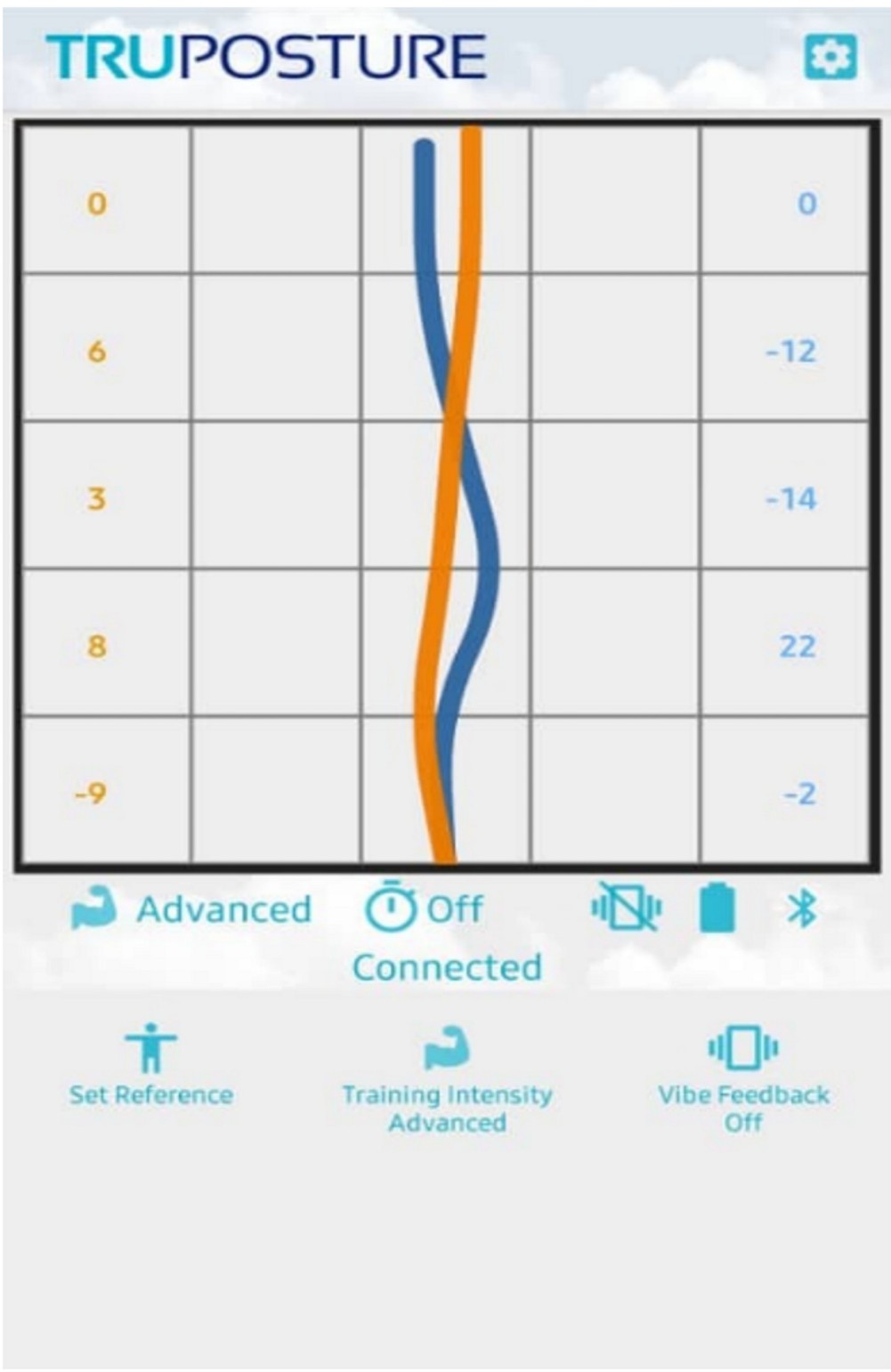

**Fig 2. TruPosture mobile apps interface.**

elbows are held easily on the holds without exceeding or over-expanding the elbows. Elbows are flexed, and lower arms are parallel to the ground. Furthermore, the legs are held near the fuel tank while relaxing the hip and pelvis. This posture was set as a spinal reference posture for riding, and the angle for each sensor was recorded. However, this is the real problem for the riders, as they could not maintain the ideal posture in medium- and long-distance trips. Thus, the subjects were asked to ride a motorcycle for only 20 minutes. This is because the average commute time of traffic police riders for one task is less than 30 minutes. Moreover, According to Deros et al. [19], 82.5% of the discomfort variance divergence is accounted for after being seated for 20 minutes and it takes less than five minutes in postural changes without back support. The riding posture was assessed by the same person (an ergonomist) in the pre-test-posttest session.

This study was conducted at the MEX highway route (the route between the commercial centre of Kuala Lumpur to the Federal Administrative capital of Putrajaya and Kuala Lumpur International Airport), with good road condition and maintenance. The route was chosen as it was commonly used in their work task as escort riders. The pre-post test results were then compared between the two groups.

## Quality control

A pre-test was conducted approximately 10% of the sample size. Two respondents who had undergone pre-test were not included in the real experimental testing as it was not to prove the superiority of the treatment but to test the procedures and processes and estimate parameters for the main trial sample size calculation. Other than that, the pre-test could ensure that all the instruments were working properly and in good order. The other purpose of this study was to familiarise the researcher with the placement of sensors of Truposture smart shirt to avoid any mistakes during the data collection process.

To ensure a good quality control, the supplier had demonstrated the handling of Truposture smart shirt to the researcher and the possible errors that might occur. The measurement procedure used was obtained from the Truposture smart shirt Manual Book. The respondents must ensure that the size of the shirt was fitted with their body to make sure the sensor placed aligned their spinal correctly. Then, the respondents need to ride a motorcycle for 20 minutes. Each test was taken around 10:00 am to 10:20 am, to ensure consistency of traffic conditions. The measurement was taken for 20 minutes because the average time taken by traffic police riders in riding motorcycle was approximately 20 minute per task.

## Statistical analysis

The collected data from the questionnaire and measurement were analysed using the IBM SPSS Software (Version 26). The Shapiro-wilk test was used in this study to determine the normality of data distribution for variables. The normality of data distribution was assumed when the p-value was more than 0.05. The present study found that the data variables were not normally distributed. Thus, a non-parametric test was used in this study. The data collected were analysed using the Mann-Whitney test and Wilcoxon signed-ranked test to verify a statistically significant difference between and within the control and experimental groups. The dependent variable of this study was the spinal angle ($0^{th}$ and $20^{th}$ minutes). The measurement of the spinal angle was done by using Truposture smart shirt. The study was conducted using a 95% confidence level, 80% of power, and the results of $p \leq 0.05$ were considered significant.

### Ethics approval

This study was submitted and approved by the Ethics Committee, Universiti Putra Malaysia (reference number: UPM/TNCPI/RMC/JKEUPM/1.4.18.2 (JKEUPM)). Permission from the subjects of this study was obtained with their written consents before the study was conducted. Privacy of the information and confidentiality of the subjects were and are always protected.

## Results

### Subjects

In total, 24 traffic police riders were included in the study, and there were no drops out. All subjects enrolled completed the study protocol as planned. The subjects' demographic data from both the control group and the experimental group are tabulated in Table 1; no significant differences were observed between the two groups.

### Distribution of five different sensors for spinal posture angle

The mean spinal posture angle in each sensor for the control and experimental groups is presented in Fig 3. The graph showed that the experimental group (post-test) values are successfully maintained with minimal changes to the spinal posture angle compared to the other groups. The positive and negative values of angle indicate the position of the spinal, which is forward and backwards, respectively.

Meanwhile, the line graph in (Fig 4a–4e) show the trend comparison of spinal posture angle deviation throughout the 20 minutes riding session. The line graph compares the level of spinal angle deviation in four groups (control pre-test, experimental pre-test, control post-test, and experimental post-test). Based on the graph line comparison, a deviation of spinal angle increased in each sensor, and Sensor 1 experienced the greatest deviation from the other sensors. From the graph, a deviation of spinal angle on each group shows an upward trend. However, the spinal angle deviation in the experimental (post-test) group is always at a lower level than the other groups. It showed that the experimental group intervention has a huge improvement in maintaining posture, which only involves slight spinal angle deviation changes from the spinal reference angle throughout the 20 minutes ride. The details of the spinal posture angle profile throughout 20 minutes are available as supporting information; S1.

**Table 1. Distribution of socio-demographic and occupational profiles between the control group and experimental group (N = 24).**

| Variable | Control group (n = 12) | Experimental group (n = 12) |
|---|---|---|
| Age (years) | | |
| Mean±SD | 30.25±5.72 | 28.92±3.78 |
| Range | 22–38 | 24–37 |
| BMI (kg/m²) | | |
| Mean±SD | 24.29±2.26 | 24.04±3.17 |
| Range | 20.55–29.05 | 19.38–29.1 |
| Working hours per day (hours) | | |
| Mean±SD | 9.17±1.75 | 9.50±1.88 |
| Range | 8.0–14.0 | 8.0–13.0 |
| Riding in a day (hours) | | |
| Mean±SD | 4.50±1.64 | 5.08±1.97 |
| Range | 2.0–8.0 | 3.0–8.0 |
| Year of service | 5.67±3.28 | 5.84±2.98 |

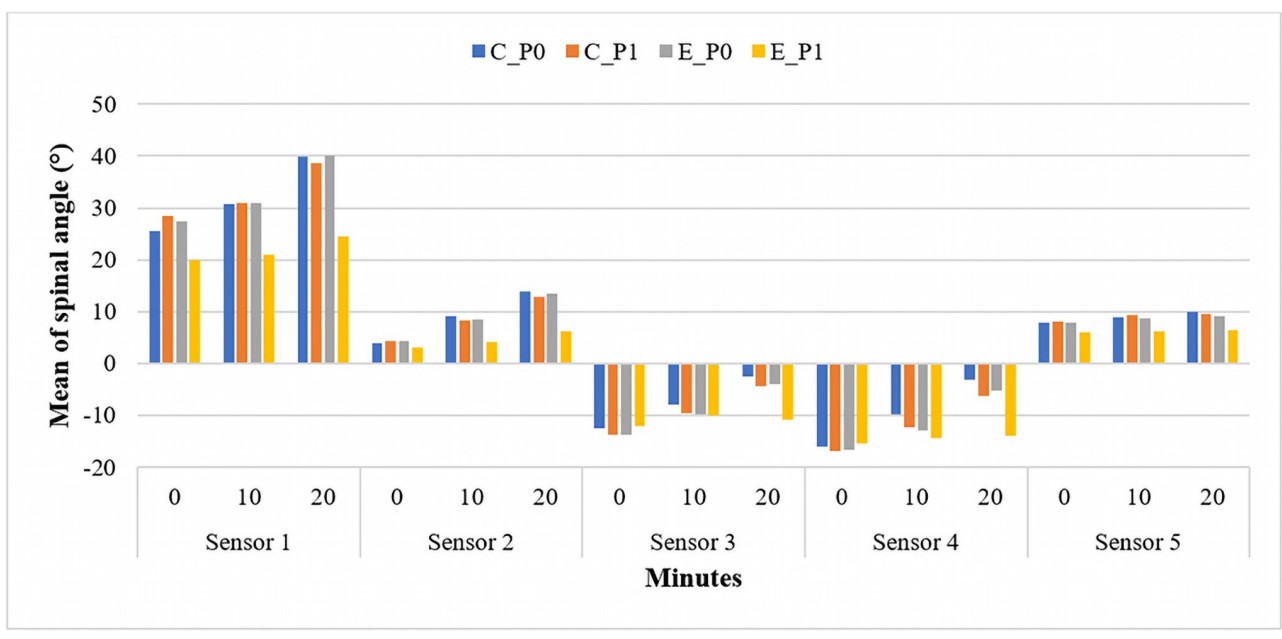

**Fig 3. Mean spinal posture angle changes throughout 20 minutes riding for pre-test and post-test study.** C_$P_0$: A control pre-test group; C_$P_1$: A control post-test; E_$P_0$: Experimental pre-test; E_$P_1$: Experimental post test.

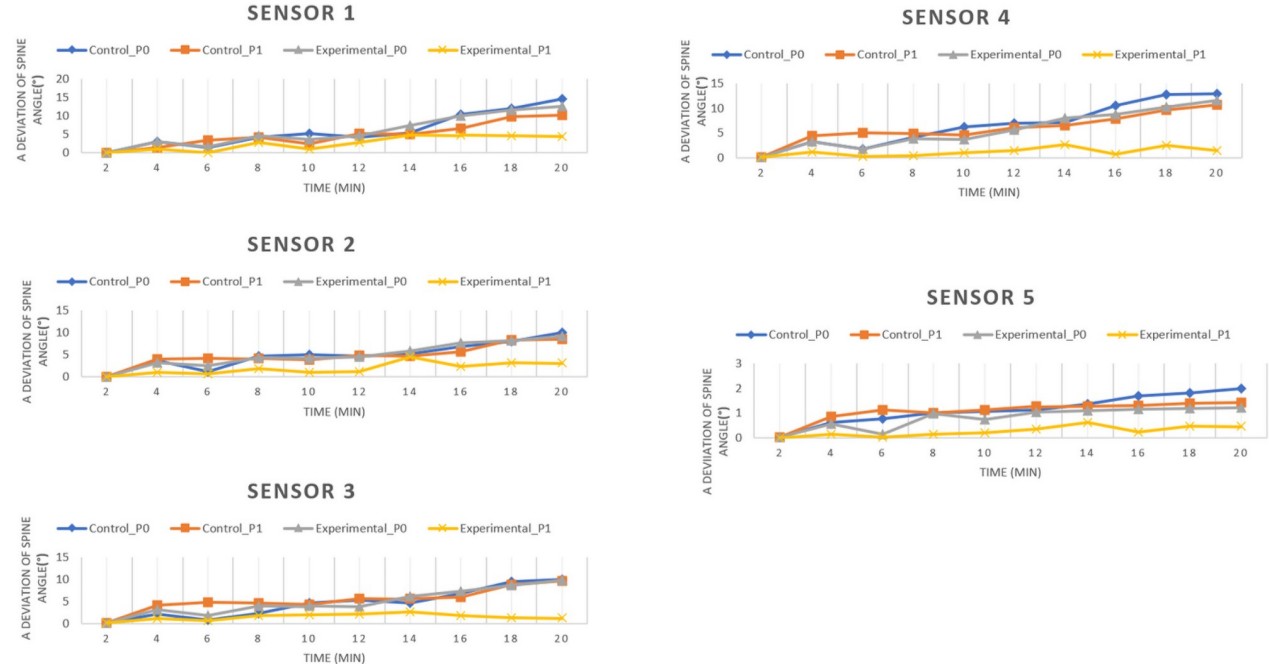

**Fig 4.** (a–e). Deviation of spinal posture angle throughout 20 minutes (N = 24).

## Comparison of five different sensors

Table 2 compares the spinal angle (0th and 20th minutes) between pre-test and post-test study for five different sensors within two groups. As depicted in this table, a Wilcoxon signed-rank test revealed a statistically significant difference between pre-test and post-test session in all sensors at the 0th and 20th minutes in the experimental group (p<0.05). Meanwhile, no statistically significant difference is observed among the control group in all five sensors (p>0.05). The results obtained from a Mann-Whitney test showed that there was no statistically significant difference in spinal angle between the control group and experimental group in the pre-test session as shown in Table 3. However, there were statistically significant differences in the spinal angle between control and experimental groups for all sensor (p<0.05) throughout 20 minutes of a motorcycle ride.

**Table 2. A median angle of five different sensors between pre-test and post-test studies within two groups.**

| Sensor | Minute | Control | | | | Experimental | | | |
|---|---|---|---|---|---|---|---|---|---|
| | | Median (IQR) | | Z-statistic | p-value | Median (IQR) | | Z-statistic | p-value |
| | | Pre- | Post- | | | Pre- | Post- | | |
| 1 | 0 | 25.5 (2.54) | 26.4 (2.57) | -1.193 | 0.471 | 25.7 (2.75) | 20.5 (3.75) | -3.065 | 0.002** |
| | 20 | 38.5 (2.73) | 38.6 (1.76) | -1.021 | 0.307 | 40.0 (1.75) | 24.5 (1.00) | -3.070 | 0.002** |
| 2 | 0 | 4.0 (2.50) | 4.71 (2.00) | -1.615 | 0.160 | 4.8 (1.75) | 3.71 (1.00) | -2.914 | 0.004** |
| | 20 | 12.5 (2.70) | 12.0 (3.00) | -1.725 | 0.084 | 12.6 (0.32) | 6.23 (0.98) | -3.061 | 0.002** |
| 3 | 0 | -12.0 (2.75) | -12.0 (3.00) | -0.751 | 0.453 | -13.0 (2.69) | -11.0 (1.75) | -2.754 | 0.006** |
| | 20 | -3.0 (1.00) | -3.5 (1.00) | -1.308 | 0.191 | -4.00 (1.75) | -10.5 (1.00) | -3.077 | 0.002** |
| 4 | 0 | -16.0 (2.50) | -16.0 (1.75) | -0.206 | 0.837 | -16.0 (2.00) | -15.0 (1.50) | -2.263 | 0.024* |
| | 20 | -4.0 (2.00) | -4.5 (2.75) | -1.083 | 0.279 | -5.0 (1.75) | -12.0 (2.00) | -3.074 | 0.002* |
| 5 | 0 | 7.0 (1.75) | 8.0 (2.75) | -1.533 | 0.125 | 8.0 (1.75) | 6.0 (1.75) | -2.090 | 0.037* |
| | 20 | 9.0 (1.00) | 10.0 (2.0) | -1.408 | 0.159 | 9.5 (1.00) | 6.5 (1.00) | -2.961 | 0.003** |

*p-value is significant at p<0.05.
**p-value is significant at p<0.01.

**Table 3. The comparison of the median spinal angle between control and experimental groups.**

| Sensor | Minute | Pre-test | | | | Post-test | | | |
|---|---|---|---|---|---|---|---|---|---|
| | | Median (IQR) | | Z-statistic | p-value | Median (IQR) | | Z-statistic | p-value |
| | | Control | Experimental | | | Control | Experimental | | |
| 1 | 0 | 25.5 (2.54) | 25.7 (2.75) | -1.172 | 0.241 | 26.4 (2.57) | 20.5 (3.75) | -4.129 | 0.004** |
| | 20 | 38.5 (2.73) | 40.0 (1.75) | -1.995 | 0.053 | 38.6 (1.76) | 24.5 (1.00) | -4.225 | 0.001** |
| 2 | 0 | 4.0 (2.50) | 4.8 (1.75) | -1.505 | 0.143 | 4.71 (2.00) | 3.71 (1.00) | -2.209 | 0.027* |
| | 20 | 12.5 (2.70) | 12.6 (0.32) | -1.879 | 0.064 | 12.0 (3.00) | 6.23 (0.98) | -4.197 | 0.002** |
| 3 | 0 | -12.0 (2.75) | -13.0 (2.69) | -1.318 | 0.187 | -12.0 (3.00) | -11.0 (1.75) | -2.385 | 0.016* |
| | 20 | -3.0 (1.00) | -4.00 (1.75) | -1.792 | 0.079 | -3.5 (1.00) | -10.5 (1.00) | -4.227 | 0.001** |
| 4 | 0 | -16.0 (2.50) | -16.0 (2.00) | -0.412 | 0.681 | -16.0 (1.75) | -15.0 (1.50) | -2.923 | 0.003** |
| | 20 | -4.0 (2.00) | -5.0 (1.75) | -1.869 | 0.073 | -4.5 (2.75) | -12.0 (2.00) | -4.189 | 0.001** |
| 5 | 0 | 7.0 (1.75) | 7.0 (1.75) | -1.285 | 0.199 | 8.0 (2.75) | 6.0 (1.75) | -2.359 | 0.018* |
| | 20 | 9.0 (1.00) | 9.5 (1.00) | -1.246 | 0.810 | 10.0 (2.0) | 6.5 (1.00) | -4.078 | 0.001** |

*p-value is significant at p<0.05.
**p-value is significant at p<0.01.

## Discussion

The present study found that the existing seat contributes to the huge changes in Sensor 1 (Thoracic 1) throughout 20 minutes of riding for both groups. This could be explained when the riders applied a slumped posture at the end of the 20 minutes of riding due to muscle fatigue while adopting an upright posture without back support. Kwon et al. [23] explained that the thoracic angle is usually related to the outcome of a slumped posture. These findings were supported by Ma'arof et al. [24], who reported that an upright riding posture involved higher muscular activity compared to the forward-lean posture, which makes riders muscle less comfortable, leading to fatigue. Shoulder stiffness is frequently encountered by motorcyclists, including young and healthy riders [13]. Therefore, officers using existing seat proved that they could not maintain an ideal and good posture while riding a motorcycle, indicating the leading cause of concern for this problem.

The application of lumbar support with a built-in massager system showed a lower deviation angle in the trend compared to the group without intervention. This happened because the upright posture without back support (exiting seat) leads to the rider's unnatural spinal curve. This can be explained by the fact that this prototype helps correct the spinal posture (upright) of motorcyclists better than the absence of lumbar support. These findings are parallel with the previous research that reported the lumbar support on a motorcycle seat was capabl of providing good posture and reduce muscle discomfort of riders during the riding process [8]. This is supported by Ceunen et al. [25], who stated that the upright sitting posture with lumbar support is efficient to change pelvis and spinal structure to their natural curve during sitting, which reduces the load on the ischial tuberosity and lower spinal, reduce muscular activity, maintained lumbar lordosis; thus, potentially reduce the risk of developing low back pain. Alyami and Albarrati [26] also found that the workstation would be safer when the posture's biomechanical risk could be reduced by applying ergonomic rules in design that would support healthy (ideal) body posture.

A statistically significant difference was noted throughout this research in all sensors at the $0^{th}$, and $20^{th}$ minutes ($p<0.05$) between pre-test and post-test measurement within the experimental group. However, there was no statistically significant difference in spinal angle within the control group. This proved that the intervention of the prototype seat in the current study could effectively maintain spinal posture in an upright riding position throughout 20 minutes riding session. The value of spinal angle deviation between the pre-test and post-test in the experimental group throughout 20 minutes of riding also showed that the slumped spinal posture could be prevented with lumbar support and a built-in massager system.

The analyses between-group differences also found that there was a significant difference in all spinal angle between control and experimental groups throughout 20 minutes of a motorcycle ride. The findings in the current study proved that the prototype seat provides a positive effect on the spinal posture of motorcyclists in adjusted and maintaining their spinal posture which can serve as a channel for the distribution of the force as well as static loading by the body. In this case, the intradiscal pressure could be reduced and the supporting back muscle could be enhanced [27] as shown in Fig 5.

Another crucial finding in this study involved applying the massager system (4 minutes off to 1 minute on), whereby the results of the experimental group showed relatively not much deviation compared to the other groups. The value of spinal angle deviation in the experimental group was reduced after every four minutes. This could be possibly due to the massager in the lumbar system improves spinal posture from an awkward posture (slump) over time. These results proved the theory that the massager in the lumbar area could reduce muscle fatigue, increase blood flow and oxygenation, and maintain body posture effectively [8–10].

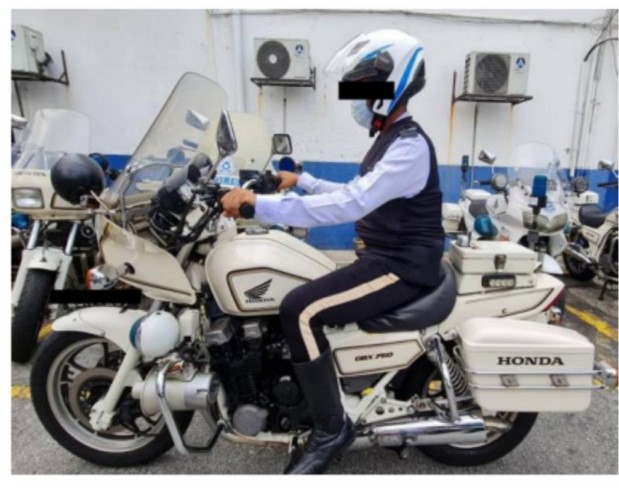 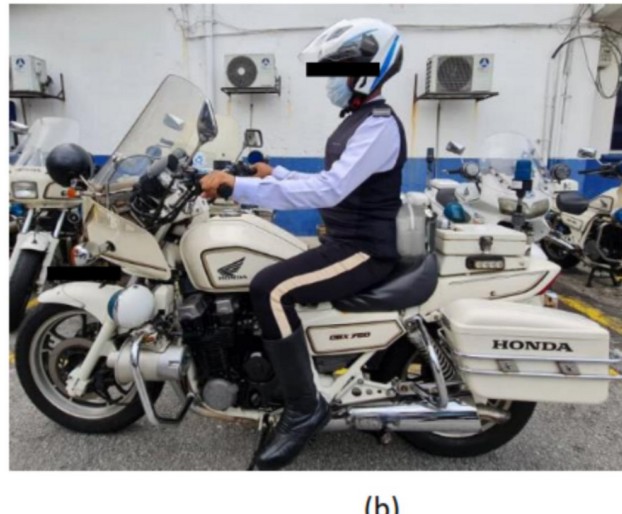

(a)                                                                (b)

**Fig 5. Posture after 20 minutes of the riding session.** (a) Existing seat. (b) Prototype seat (lumbar support with build-in massager system).

Contrarily, Tanaka et al. [28] reported differences that suggested the lumbar massage failed to exert significant changes in any electromyographic (EMG) measurement. Besides that, Van Poppel et al. [29] reported that lumbar support was not useful in preventing low back pain. The outcome of this study contradicted the study by Kolich et al. [18], where they pointed out that one minute of lumbar massage in every five minutes of the driving session has a beneficial effect on the lower back muscle activity. A study conducted by Franz et al. [30] suggested that lumbar support usage with lightweight massager in automobile seats has been reported to decrease muscle discomfort during the driving process and improve lumbar lordosis.

There were several limitations in the present study. Only male traffic police riders were recruited. Thus, it cannot be used to generalise the whole population. A future study might include female traffic police riders and other occupations such as food delivery. Even though we found a positive outcome of this prototype in maintaining spinal posture, since this study was conducted in an on-the-road setting, there was a limited objective measurement that can be done such as the measurement of muscle fatigue using EMG electrode (clinical testing) due to environmental factors such as road condition and vibration. This study was unable to blind the respondents to the condition that they were exposed to, which might impact the posture riding in this study. This is because traffic police riders were aware of which seat they were using, and as such, there was no way to prevent a bias toward the prototype seat. However, it is assumed that any bias towards the motorcycle seat prototype would disappear through the riding duration if the seat did not truly adjust and maintain riding posture throughout the riding process. Due to time constraints (pandemic outbreak) and limited resources such as budget, it was not possible to include all aspects of the problem with a bigger sample size in this research. Besides that, there was only one prototype available for this research.

## Conclusion

In summary, the present study proved that lumbar support with a built-in massager system successfully maintains and improves the spinal posture angle ergonomically throughout 20 minutes of riding. Although there is a positive outcome in this research, there is still a lack of

evidence (i.e. muscle fatigue, discomfort, muscle activity) for this study. Hence, further study in the artificial laboratory and clinical settings as well as virtual testing is needed to support these findings.

## Supporting information

**S1 Table. Spinal angle profile throughout 20 minutes (N = 24).**
(DOCX)

## Acknowledgments

The author would like to thank all of the people who were involved in this study especially the Royal Malaysian Police (RMP).

## Author Contributions

**Conceptualization:** Nur Athirah Diyana Mohammad Yusof, Kulanthayan K. C. Mani.

**Data curation:** Vivien How, Putri Anis Syahira Mohamad Jamil.

**Formal analysis:** Putri Anis Syahira Mohamad Jamil, Hassan Sadeghi Naeini.

**Funding acquisition:** Karmegam Karupiah.

**Investigation:** Nur Athirah Diyana Mohammad Yusof, Putri Anis Syahira Mohamad Jamil.

**Methodology:** Nur Athirah Diyana Mohammad Yusof, Putri Anis Syahira Mohamad Jamil.

**Project administration:** Nur Athirah Diyana Mohammad Yusof, Karmegam Karupiah.

**Resources:** Irniza Rasdi, Sivasankar Sambasivam, Putri Anis Syahira Mohamad Jamil, Dayana Hazwani Mohd Suadi Nata.

**Software:** Irniza Rasdi, Dayana Hazwani Mohd Suadi Nata.

**Supervision:** Karmegam Karupiah, Shamsul Bahri Mohd Tamrin, Vivien How, Kulanthayan K. C. Mani, Hassan Sadeghi Naeini.

**Validation:** Nur Athirah Diyana Mohammad Yusof, Shamsul Bahri Mohd Tamrin, Vivien How, Sivasankar Sambasivam, Putri Anis Syahira Mohamad Jamil.

**Visualization:** Shamsul Bahri Mohd Tamrin.

**Writing – original draft:** Nur Athirah Diyana Mohammad Yusof.

**Writing – review & editing:** Nur Athirah Diyana Mohammad Yusof, Sivasankar Sambasivam.

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
