## [Decision Letter · Decision Letter 0]

27 May 2021

PONE-D-21-13337

Assessment of spinal angle profiles among high-powered traffic police motorcycle riders

PLOS ONE

Dear Dr. Karuppiah,

Thank you for submitting your manuscript to PLOS ONE. After careful consideration, we feel that it has merit but does not fully meet PLOS ONE’s publication criteria as it currently stands. Therefore, we invite you to submit a revised version of the manuscript that addresses the points raised during the review process.

Please consider all the comments of all reviewers including the comments of Reviewer 3

We look forward to receiving your revised manuscript.

Kind regards,

Ahmed Mancy Mosa, Ph.D.

Academic Editor

PLOS ONE

Journal Requirements:

Reviewers' comments:

Reviewer's Responses to Questions

**Comments to the Author**

1. Is the manuscript technically sound, and do the data support the conclusions?

Reviewer #1: Partly

Reviewer #2: No

Reviewer #3: No

Reviewer #4: No

Reviewer #5: Yes

2. Has the statistical analysis been performed appropriately and rigorously? 

Reviewer #1: No

Reviewer #2: No

Reviewer #3: No

Reviewer #4: No

Reviewer #5: Yes

3. Have the authors made all data underlying the findings in their manuscript fully available?

Reviewer #1: Yes

Reviewer #2: Yes

Reviewer #3: Yes

Reviewer #4: No

Reviewer #5: Yes

4. Is the manuscript presented in an intelligible fashion and written in standard English?

Reviewer #1: No

Reviewer #2: Yes

Reviewer #3: No

Reviewer #4: No

Reviewer #5: Yes

5. Review Comments to the Author

Reviewer #1: Dear Authors,

This is an interesting research area. While the study has some merits to consider for publication several clarifications are needed to assist in the decision. My comments and questions are as follow:

1. What is the new contribution of this study? What was previously known about the ergonomic function and effect of the prototype seat, and how will this study add to the existing knowledge? Why is it important to measure the spinal angle profile?

2. The problem statement highlighted in the introduction does not match the aim of the study which is an experimental study comparing between an intervention and control group.

3. Please clarify whether there is any sample size calculation undertaken prior to the study, and provide the relevant references for the sample size calculation. What was the power of the study based on the sample size calculation?

4. Please provide adequate references for the prototype and the TruPosture app.

5. Please provide a detailed explanation on the recruitment process, and the allocation to control and intervention group.

6. What was the dependent variable in the statistical analysis. please provide the operational definition of the dependent variable.

7. Please provide the general description of your study participants in the results section, and describe the characteristics between the experimental and control group to show that they are comparable.

8. Do you expect any confounding factors in your final analysis to look at the difference in outcome between the intervention and control group? What are the appropriate statistical tests to be used to control for the confounders, if there is any?

9. please explain clearly the within and between groups comparison in the final analysis of the study outcome.

10. Please align the discussion with the aim of the manuscript. What is it that you plan to highlight? is it about the

usability of the TruPosture smart Shirt in measuring posture, or it is about showing the effect of the prototype seat on posture of the police riders?

11. What is the study limitation?

12. Please revise the manuscript title to reflect on the aim and experimental design.

Reviewer #2: 1. SUMMARY OF THE RESEARCH AND OVERALL IMPRESSION –

The manuscript is well written with concise language, but I have major concerns on the methodology that prevent me to endorse acceptance at current stage. I see that sound methodology is lacking and that compromise the data analyses and could have led to potentially wrong results and misleading conclusion. Recommended course of action is to re-look at the methods and try to conform to standard reporting guidelines such as CONSORT guideline. The authors can expand the study sample size, with proper calculation and reference to previous study to ensure its adequate to draw to solid conclusion. Current study can serve as preliminary or pilot study to come up with better technical standard of experimental methodology and sufficient description on the details.

2. DISCUSSION OF SPECIFIC AREAS FOR IMPROVEMENT –

a) Methodology (major issues)

There are several issues with the methodology and analysis that need to be clarified, address and described in great detail. The methodology section should be expanded and clarified to support the conclusion and validate the findings. As the methods section is lacking, it might be premature to draw sound and valid conclusion.

• There is no mention on the specific study design and how the randomisation is done (authors only mentioned randomisation once in Figure 2). Is this a randomised controlled trial? Why is it not mentioned in the title? Has it been registered in any trial’s registry?

• There is no mention of how the sample size is calculated. How is the sample size being determined? Is the sample size adequate to represent the study population? Is the sample size adequate to meet the assumption of statistical analyses and come to conclusion?

• Eligibility criteria is not clear (authors only mention no history of MSD and low back pain, but what about baseline age, height, weight, BMI, fat percentage, gender, ethnicity, years of service, and other factors that might influence the spinal angle profile?).

• There is no mention on how the recruitment is done. Is it convenience? Can it be representative of the study population? Who enrolled the participants?

• How is the allocation to the control and intervention group being done? Is there any blinding involved? What are the methods used to do randomisation and type of randomisation (when allocating participants to control or intervention group)? The authors did not mention who did the randomisation and who assigned the participants interventions.

• There is no mention of the time period of data collection defining recruitment, pre and post-test. How long is the gap between pre and post-test? What specific time of the day the experiment being conducted? (is the timing and duration of work on that particular experiment day influence the spinal angle profiles?) What is the justification for 20 minutes duration of the experimental riding session? (authors mention riders spend at least 5 hours per day on motorbikes).

• The author did not mention anything on the instrument validation and calibration. How are the sensors being placed? Is it done by the same person for both control and intervention groups? Has the apps being tested and validated before? How accurate is the reading?

• The author did not mention whether both groups are being assessed by the same person (e.g. posture for riding). If it’s not done by the same person, do the researchers take into account inter-rater reliability?

b) Discussion

There is no mention or any discussion on the experiment’s limitations in the discussion part. The authors did not discuss potential source bias and confounder, threat to validity that might compromise the findings.

c) Results

• The authors did not report the sociodemographic and baseline characteristics of both control and experimental group. How does the researcher ensure that both groups are similar at baseline? How can the researcher then conclude that the outcome is due to their intervention, rather than existing differences? Are there any additional methods of analyses such as subgroup analyses to account for confounder or differences in the baseline?

• There is no justification on why non-parametric statistical analyses was chosen. What are the assumptions and limitation of the analysis? Have all the criteria being met?

• Too much data being presented that can be summarised in a sentence or in simpler table. For example, Table 1 presents a range of minimum and maximum value for each reading throughout 20 minutes – can be summarised by providing the mean/median value. Easier to interpret and understand, rather than the reader have to go thru each range.

d) Others

• Figure 1 – no legend to help reader understand the figure. Which line belongs to control and intervention group?

• Figure 2 – suggest authors to follow CONSORT flow diagram format (more details)

3. OTHER POINTS

Authors have provided adequate literature review to justify the problem statement, significance of the study and burden of the disease. The authors also adequately described previous research and gap of the study.

The data presented has potential to be published if its extended and properly developed. Current study can serve as preliminary findings or pilot to come up with better protocols and larger sample size to test the same hypotheses and draw more concrete results and conclusion.

Reviewer #3: Thank you for the opportunity to review your manuscript entitled “Assessment of spinal angle profiles among high-powered traffic police motorcycle riders”. The topic is interesting; however, I believe the study was not appropriately designed to address the research question. The analysis and reporting should not lead to the conclusion made by the authors. The introduction needs to be strengthened. I have highlighted some points that I believe would improve the quality of the manuscript.

Abstract

Line 30: the purpose of the study described in the abstract does not match with what was done in the methods. This was an intervention study and I suggest reformatting the purpose of the study.

Line 40: Are we really interested in a pre-test VS post-test analysis (within group analysis)? Reporting the between group comparison is more insightful…

Introduction

Provide some statistics of MSD/low back in the population

Line 51: provide a full meaning of MSD

Line 59: I believe it was meant to be “been” instead of “seen”

Line 61: has instead of have

Line 67: delete “the”

Line 81: The purpose of the study needs to be reformulated

The introduction lacks content. The consequences of increased lumbar lordosis are not fully described. There are some grammatical errors that need to be corrected.

Materials and Methods

More information on the eligibility criteria (inclusion and exclusion criteria) is needed

Any reference for the TruePosture mobile app? Has this app been used before in any study? What are the validity and reliability properties of the app?

How were the groups defined? Were the participants randomly assigned to the groups? What are the baseline characteristics of the control and experimental groups?

Why did the authors not perform a between-group analysis? That comparison is more interesting than all those pre-test vs post-tests performed.

Results

Data on the participants missing

The data reported does not indicate whether the spinal change pattern in the intervention group is superior to the control group.

Discussion and conclusion

Not sustained by the analysis and results presented

Reviewer #4: Abstract

Background

Context and research gap were not indicated

Materials and Methods

Type of study, sampling method and data collection method were not specified

Major statistical analysis was not stated

Main body

Materials and Methods

It has major methodological defect.

Type of study, sampling method and data collection method were not specified

Gold standard of experimental study was not succinctly stated.

Outcome was not assessed

Major statistical analysis was not indicated.

Why Wilcoxon signed rank test and median was used?

Reviewer #5: Line 51: Please indicate the full meaning of MSD when been used for the first time in write up

Line 178: Consider replacing "respondents" with "subjects" which best suits the study and its concept

Please give reasons for the sample size choice and indicate the exclusion and inclusion criteria for participating in the study

Line 243: "The value of spinal angle deviation between the pre-test and post-test ....." Indicate the angle of deviation to make your point clear

There are some typographical errors indicated in the attached document. Please revise them accordingly

6. PLOS authors have the option to publish the peer review history of their article (what does this mean?). If published, this will include your full peer review and any attached files.

Reviewer #1: No

Reviewer #2: No

Reviewer #3: No

Reviewer #4: No

Reviewer #5: No

---

## [Author Response · Author response to Decision Letter 0]

4 Jun 2021

Dear Dr Ahmed Mancy Mosa (Academic Editor), 

Thank you for giving me the opportunity to submit a revised draft of my manuscript titled Assessment of spinal angle profiles among high-powered traffic police motorcycle riders. We appreciate the time and effort that you and the reviewers have dedicated to providing your valuable feedback on my manuscript. We are grateful to the reviewers for their insightful comments on our paper. We have been able to incorporate changes to reflect most of the suggestions provided by the reviewers. We have highlighted the revisions within the manuscript. The line mention in the author response.

Here is a point-by-point response to the reviewers’ comments and concerns.

 Reviewer 1 

1. What is the new contribution of this study? 

 Thank you for pointing this out. The new contribution of this study is:

• The application of the method (Truposture smart shirt) in spinal profile measurement was evaluated in an occupational setting (on-the-road) which riding in a real working condition was applied.

• In motorcycling industry, new invention for an ergonomic motorcycle seat had been developed and tested.

2. What was previously known about the ergonomic function and effect of the prototype seat, and how will this study add to the existing knowledge? Why is it important to measure the spinal angle profile? 

We agree with this comment. Therefore, the relevant justification has been made in the introduction section (line 92- 99). 

2. The problem statement highlighted in the introduction does not match the aim of the study which is an experimental study comparing between an intervention and control group. 

We agree with this comment and has made changes accordingly in introduction section (line 94-96).

3. Please clarify whether there is any sample size calculation undertaken prior to the study, and provide the relevant references for the sample size calculation. What was the power of the study based on the sample size calculation? You have raised an important point here. 

We agree with this comment and the sample size calculation has been added in sample size section (line 122-129).

4. Please provide adequate references for the prototype and the TruPosture app. 

Adequate references for the prototype and Truposture app have made (line 186-189).

5. Please provide a detailed explanation on the recruitment process, and the allocation to control and intervention group. 

Thank you for pointing this out. The recruitment process has been explained accordingly (line 131-158).

6. What was the dependent variable in the statistical analysis. please provide the operational definition of the dependent variable. 

The dependent variable and the operational definition have been added in the statistical analysis as suggested (line 223-225).

7. Please provide the general description of your study participants in the results section, and describe the characteristics between the experimental and control group to show that they are comparable. 

Thank you for pointing this out. The general description of the participants has been discussed and tabulated in Table 1 (line 234-242).

8. Do you expect any confounding factors in your final analysis to look at the difference in outcome between the intervention and control group? What are the appropriate statistical tests to be used to control for the confounders, if there is any? 

The confounding factors in this study was expected with the difference age, gender, BMI and present of low back pain. However, these criteria were controlled based on inclusion and exclusion criteria during sampling (line 144-121). 

9. please explain clearly the within and between groups comparison in the final analysis of the study outcome. 

Thank you for pointing this out. We agree with this comment and has made changes accordingly in results and discussion section (line 316-330).

10. Please align the discussion with the aim of the manuscript. What is it that you plan to highlight? is it about the usability of the TruPosture smart Shirt in measuring posture, or it is about showing the effect of the prototype seat on posture of the police riders? 

Agree. We have, accordingly, revised the discussion part and removed the unrelated part.

11. What is the study limitation? 

Thank you for pointing this out. We have added the study limitation accordingly in the discussion section (line 350-359).

12. Please revise the manuscript title to reflect on the aim and experimental design. 

We agree with this comment. Therefore, we have revised and changed the manuscript title into “Effectiveness of lumbar support with built-in massager system on spinal angle profiles among high-powered traffic police motorcycle riders: A randomised controlled trial”

 Reviewer 2 

1. • There is no mention on the specific study design and how the randomisation is done (authors only mentioned randomisation once in Figure 2). Is this a randomised controlled trial? Why is it not mentioned in the title? Has it been registered in any trial’s registry? 

Thank you for the input. Yes, it is randomised controlled trial. However, we apologise for overlooking this matter. Thus, the title and study design have been revised as suggested (line 108-112). 

It has not been registered in any trial’s registry. We are sorry for this and hope for your understanding. Nevertheless, the sampling method is approved and supported by the Ethical Committee of Universiti Putra Malaysia after consulting their expertise.

2. • There is no mention of how the sample size is calculated. How is the sample size being determined? Is the sample size adequate to represent the study population? Is the sample size adequate to meet the assumption of statistical analyses and come to conclusion? 

You have raised an important point here. We agree with this comment and the sample size calculation has been added (line 122-129).

3. • Eligibility criteria is not clear (authors only mention no history of MSD and low back pain, but what about baseline age, height, weight, BMI, fat percentage, gender, ethnicity, years of service, and other factors that might influence the spinal angle profile?). 

You have raised an important point here. We agree with this comment and has included relevant information and explanation accordingly (line 114-121). 

However, due to time (pandemic outbreak) and budget constraints (e.g: fat percentage analyser), we are unable to include all aspects of the problem and other factors. 

Thank you for pointing this out. For future studies, this will be a great insight and we will be sure to use it. Also, we will highlight this matter as the limitation in this study.

4. • There is no mention on how the recruitment is done. Is it convenience? Can it be representative of the study population? Who enrolled the participants? 

Thank you for pointing this out. The participant recruitment has been included and explained accordingly (131-158).

5. • How is the allocation to the control and intervention group being done? Is there any blinding involved? What are the methods used to do randomisation and type of randomisation (when allocating participants to control or intervention group)? The authors did not mention who did the randomisation and who assigned the participants interventions. 

Thank you for pointing this out. The participant recruitment has been included and explained accordingly (131-158).

6. There is no mention of the time period of data collection defining recruitment, pre and post-test. How long is the gap between pre and post-test? 

Thank you for pointing this out. The participant recruitment has been included and explained accordingly (line 144).

7. What specific time of the day the experiment being conducted? (is the timing and duration of work on that particular experiment day influence the spinal angle profiles?) What is the justification for 20 minutes duration of the experimental riding session? (authors mention riders spend at least 5 hours per day on motorbikes). You have raised an important point here. 

We agree with this comment and have included the relevant information and justification in data collection section (line 205-209).

8. • The author did not mention anything on the instrument validation and calibration. How are the sensors being placed? Is it done by the same person for both control and intervention groups? Has the apps being tested and validated before? How accurate is the reading? 

We agree with this comment. Therefore, we have discussed this information in TruPosture section (line 186-192).

9. • The author did not mention whether both groups are being assessed by the same person (e.g. posture for riding). If it’s not done by the same person, do the researchers take into account inter-rater reliability?

You have raised an important point here. Thus, the relevant information has been added accordingly (line 209-210).

10. b) Discussion

There is no mention or any discussion on the experiment’s limitations in the discussion part. The authors did not discuss potential source bias and confounder, threat to validity that might compromise the findings.

Thank you for pointing this out. We have added the study limitation accordingly (line 350-364).

11. c) Results

• The authors did not report the sociodemographic and baseline characteristics of both control and experimental group. How does the researcher ensure that both groups are similar at baseline? How can the researcher then conclude that the outcome is due to their intervention, rather than existing differences? Are there any additional methods of analyses such as subgroup analyses to account for confounder or differences in the baseline?

Thank you for pointing this out. The general description of the participants has been discussed and tabulated in Table 1 (line 234-242).

12. • There is no justification on why non-parametric statistical analyses was chosen. What are the assumptions and limitation of the analysis? Have all the criteria being met? 

We agree with this comment and has made changes accordingly in statistical analysis section (line 218-223).

13. • Too much data being presented that can be summarised in a sentence or in simpler table. For example, Table 1 presents a range of minimum and maximum value for each reading throughout 20 minutes – can be summarised by providing the mean/median value. Easier to interpret and understand, rather than the reader have to go thru each range.

Agree. We have removed this table accordingly and the mean value of spinal posture angle changes throughout 20 minutes riding for both groups had already presented in Figure 3 (line 250).

15. d) Others

• Figure 1 – no legend to help reader understand the figure. Which line belongs to control and intervention group?

Actually Figure 1 is an example of Truposture mobile apps interface and not related with the control and intervention group results.

• Figure 2 – suggest authors to follow CONSORT flow diagram format (more details)

Thank you for your suggestion. Thus, Figure 2 has been changed and followed CONSORT format (Fig1: Line 146).

 Reviewer 3 

1. Abstract

Line 30: the purpose of the study described in the abstract does not match with what was done in the methods. This was an intervention study and I suggest reformatting the purpose of the study. 

The purpose of the study has been reformulated accordingly (line 30-31).

2. Line 40: Are we really interested in a pre-test VS post-test analysis (within group analysis)? Reporting the between group comparison is more insightful… 

Thank you for pointing this out. Thus, results and discussion related to comparison between groups have been added accordingly 

3. Introduction

Provide some statistics of MSD/low back in the population The statistics of MSD and low back pain among traffic police riders have been provided as suggested. (line 78-85)

4. Line 51: provide a full meaning of MSD 

The full meaning of MSD has been provided as suggested (line 50).

5. Line 59: I believe it was meant to be “been” instead of “seen” 

Thank you for the comment. The relevant word has been changed accordingly (line 104-106).

6. 

Line 61: has instead of have 

Thank you for the comment. The relevant word has been changed accordingly.

7. Line 67: delete “the” 

Thank you for the comment. The relevant word has been deleted accordingly.

8. Line 81: The purpose of the study needs to be reformulated 

The purpose of the study has been reformulated accordingly.

9. The introduction lacks content. The consequences of increased lumbar lordosis are not fully described. There are some grammatical errors that need to be corrected.

Thank you for pointing this out. The consequence of decreased lumbar lordosis has been explained accordingly (line 52-60).

10. Materials and Methods

More information on the eligibility criteria (inclusion and exclusion criteria) is needed 

You have raised an important point here. We agree with this comment and has included relevant information and explanation accordingly (114-121). 

10. Any reference for the TruePosture mobile app? Has this app been used before in any study? What are the validity and reliability properties of the app? 

Adequate references for the Truposture app have been added and discussed (line 186-189).

11. How were the groups defined? Were the participants randomly assigned to the groups? What are the baseline characteristics of the control and experimental groups? 

We agree with this comment. Thus, the relevant information has been added and explained accordingly.

12. Why did the authors not perform a between-group analysis? That comparison is more interesting than all those pre-test vs post-tests performed. 

Thank you for pointing this out. Thus, results and discussion related to the comparison between groups have been added accordingly.

13 Results

The data reported does not indicate whether the spinal change pattern in the intervention group is superior to the control group. 

Thank you for pointing this out. Thus, results and discussion related to the spinal change pattern between groups have been added accordingly.

14. Discussion and conclusion

Not sustained by the analysis and results presented 

The analysis and results presented have been revised.

Reviewer 4 

1. Context and research gap were not indicated. 

The research gap was explained in line 85.

2. Type of study, sampling method and data collection method were not specified.

It has major methodological defect.

Type of study, sampling method and data collection method were not specified

Gold standard of experimental study was not succinctly stated. 

Thank you for pointing this out. We agree with this and have incorporated your suggestion throughout the manuscript accordingly.

3. Major statistical analysis was not indicated.

Why Wilcoxon signed rank test and median was used? 

The reason the analysis was chosen has been discussed in details (line 218-223).

 Reviewer 5 

1. Reviewer #5: Line 51: Please indicate the full meaning of MSD when been used for the first time in write up. 

Thank you for pointing this out. The change has been made accordingly (line 50).

2. Line 178: Consider replacing "respondents" with "subjects" which best suits the study and its concept

 Thank you for pointing this out. The change has been made throughout the manuscript accordingly.

3. Please give reasons for the sample size choice and indicate the exclusion and inclusion criteria for participating in the study

You have raised an important point here. We agree with this comment and has included relevant information and explanation accordingly (line 114-129). 

4. Line 243: "The value of spinal angle deviation between the pre-test and post-test ....." Indicate the angle of deviation to make your point clear

Thank you for the comment. Actually, we want to highlight the effect in the experimental group. However, we have made some changes based on your suggestion

“The value of spinal angle deviation in the experimental group….”

5. There are some typographical errors indicated in the attached document. Please revise them accordingly Thank you for pointing this out. The typological error has been revised accordingly.

---

## [Decision Letter · Decision Letter 1]

12 Jul 2021

PONE-D-21-13337R1

Effectiveness of lumbar support with built-in massager system on spinal angle profiles among high-powered traffic police motorcycle riders: A randomised controlled trial

PLOS ONE

Dear Dr. Karuppiah,

Thank you for submitting your manuscript to PLOS ONE. After careful consideration, we feel that it has merit but does not fully meet PLOS ONE’s publication criteria as it currently stands. Therefore, we invite you to submit a revised version of the manuscript that addresses the points raised during the review process.

Please, carefully, consider all the comments of all reviewers

We look forward to receiving your revised manuscript.

Kind regards,

Ahmed Mancy Mosa, Ph.D.

Academic Editor

PLOS ONE

Journal Requirements:

Reviewers' comments:

Reviewer's Responses to Questions

**Comments to the Author**

1. If the authors have adequately addressed your comments raised in a previous round of review and you feel that this manuscript is now acceptable for publication, you may indicate that here to bypass the “Comments to the Author” section, enter your conflict of interest statement in the “Confidential to Editor” section, and submit your "Accept" recommendation.

Reviewer #3: All comments have been addressed

Reviewer #4: (No Response)

Reviewer #5: All comments have been addressed

2. Is the manuscript technically sound, and do the data support the conclusions?

Reviewer #3: Partly

Reviewer #4: Yes

Reviewer #5: Yes

3. Has the statistical analysis been performed appropriately and rigorously? 

Reviewer #3: No

Reviewer #4: Yes

Reviewer #5: Yes

4. Have the authors made all data underlying the findings in their manuscript fully available?

Reviewer #3: Yes

Reviewer #4: Yes

Reviewer #5: Yes

5. Is the manuscript presented in an intelligible fashion and written in standard English?

Reviewer #3: Yes

Reviewer #4: Yes

Reviewer #5: Yes

6. Review Comments to the Author

Reviewer #3: The authors appropriately revised the manuscript and made the corresponding adjustments. The amount of time devoted to this paper is evident. I have some more comments.

Additional comments:

How was the randomization performed? Any use of software or what was used for the randomization?

Table 3 is not understandable and needs to be reformatted. The formatting of this Table is fine for a within group difference analysis like the authors did for Table 2 but not for a between group difference analysis. A Table should be self-explanatory and this one is not. What does the p-value in the control column indicate? Difference between baseline data (Precont VS Preexp)? Similarly, what does p-value in the experimental column indicate? Difference between after intervention data (Postcont VS Postexp)?

I would even suggest comparing the median of changes between control and experimental groups.

Line 277: The sentence about no significant difference at baseline should be presented before the results of after intervention. This is a good sign for a between group comparison after intervention.

Reviewer #4: Abstract

Background

Context and research gap were not stated.

methods

major statistical analysis was not succinctly stated

Main body

Methods

Sampling technique was not clear

data quality control was not indicated

Major statistical analysis was not indicated

Reviewer #5: The manuscript is well written and technically sound. All the issues have been addressed by the author.

However, there are a couple of technical and grammatical errors that have been noticed:

Line 132: “ the recruitment strategy used by taken the name list of all officers”

It should rather read “ the recruitment strategy involved taking the name list of all the officers”

Line 139: “….then a main researcher was randomly assigned them …….”

It should rather read “………, then the main researcher randomly assigned them into control and experimental

Line 189: “this equipment had been tested validity….”

It should rather read “the validity of the equipment had been tested and approved……”

Other comments have been made in the attached document

7. PLOS authors have the option to publish the peer review history of their article (what does this mean?). If published, this will include your full peer review and any attached files.

Reviewer #3: **Yes: **Libak Abou

Reviewer #4: No

Reviewer #5: No

---

## [Author Response · Author response to Decision Letter 1]

15 Jul 2021

Dear Dr Ahmed Mancy Mosa (Academic Editor), 

Thank you for giving me the opportunity to submit a second revised draft of my manuscript titled effectiveness of lumbar support with built-in massager system on spinal angle profiles among high-powered traffic police motorcycle riders: A randomised controlled trial. We appreciate the time and effort that you and the reviewers have dedicated to providing your valuable feedback on my manuscript. We are grateful to the reviewers for their insightful comments on our paper. We have been able to incorporate changes to reflect most of the suggestions provided by the reviewers. We have highlighted the revisions within the manuscript. 

Here is a point-by-point response to the reviewers’ comments and concerns.

Reviewer 3

1. How was the randomization performed? Any use of software or what was used for the randomization?

 The randomization of this study was explained using the list name and Fishbowl technique (line 140-142).

2. Table 3 is not understandable and needs to be reformatted. The formatting of this Table is fine for a within group difference analysis like the authors did for Table 2 but not for a between group difference analysis. A Table should be self-explanatory and this one is not. What does the p-value in the control column indicate? Difference between baseline data (Precont VS Preexp)? Similarly, what does p-value in the experimental column indicate? Difference between after intervention data (Postcont VS Postexp)? I would even suggest comparing the median of changes between control and experimental groups. 

We are sorry for the confusion, and for our understanding we did this as mentioned in the earlier comments (i. Line 40: Are we really interested in a pre-test VS post-test analysis (within group analysis)? Reporting the between group comparison is more insightful…ii. Please explain within and between groups comparison in the analysis of the study outcome.). 

Thus, we created Table 2 for within-group analysis and Table 3 reporting between-group comparisons as suggested. The p-value in Table 2 represents a value for a significant/not significant between pre-test and post-test studies within two groups and p-value in Table 3 explain a value for a significant/not significant between control and experimental groups.

3. Line 277: The sentence about no significant difference at baseline should be presented before the results of after intervention. This is a good sign for a between group comparison after intervention. 

Thank you for your suggestion. We agree with this comment. The revision has been made on lines 293-297.

Reviewer 4

1. Background:Context and research gap were not stated. 

i. The context and research gap of traffic police riders suffered low back pain, and MSD which no research has been done on the spinal riding posture during a motorcycle ride was explained in line 72-87.

ii. The context and research gap in which there is a lack of data on the effectiveness of this intervention seat, lumbar support with a built-in massager system, in an in-field setting (on-the-road) was explained in line 96-99.

2. methods: major statistical analysis was not succinctly stated.

The major statistical analysis used in this study was the Mann-Whitney test and Wilcoxon signed-ranked test. The details of the statistical analysis were explained in line 233-243.

3. Methods: Sampling technique was not clear. 

The sampling technique was explained in Participant Recruitment (line 132-158).

4. data quality control was not indicated. 

Thank you for pointing this out. We agree with this comment. The data quality control had been added in line 216-231.

Reviewer 5 

1. Line 132: “ the recruitment strategy used by taken the name list of all officers”

It should rather read “ the recruitment strategy involved taking the name list of all the officers” 

Thank you for pointing this out. The change has been made accordingly (line 132).

2. Line 139: “….then a main researcher was randomly assigned them …….”

It should rather read “………, then the main researcher randomly assigned them into control and experimental Thank you for pointing this out. The change has been made accordingly (line 139).

3. Line 189: “this equipment had been tested validity….”

It should rather read “the validity of the equipment had been tested and approved……” 

Thank you for pointing this out. The change has been made accordingly (line 189).

4. Other comments have been made in the attached document 

Thank you for the comments. We agree with this and have incorporated your suggestion throughout the manuscript accordingly.

---

## [Decision Letter · Decision Letter 2]

6 Sep 2021

PONE-D-21-13337R2Effectiveness of lumbar support with built-in massager system on spinal angle profiles among high-powered traffic police motorcycle riders: A randomised controlled trialPLOS ONE

Dear Dr. Karuppiah,

Thank you for submitting your manuscript to PLOS ONE. After careful consideration, we feel that it has merit but does not fully meet PLOS ONE’s publication criteria as it currently stands. Therefore, we invite you to submit a revised version of the manuscript that addresses the points raised during the review process.

Please, consider all the comments

We look forward to receiving your revised manuscript.

Kind regards,

Ahmed Mancy Mosa, Ph.D.

Academic Editor

PLOS ONE

Journal Requirements:

Additional Editor Comments (if provided):

Reviewers' comments:

Reviewer's Responses to Questions

**Comments to the Author**

1. If the authors have adequately addressed your comments raised in a previous round of review and you feel that this manuscript is now acceptable for publication, you may indicate that here to bypass the “Comments to the Author” section, enter your conflict of interest statement in the “Confidential to Editor” section, and submit your "Accept" recommendation.

Reviewer #3: (No Response)

Reviewer #4: All comments have been addressed

2. Is the manuscript technically sound, and do the data support the conclusions?

Reviewer #3: Yes

Reviewer #4: Yes

3. Has the statistical analysis been performed appropriately and rigorously? 

Reviewer #3: No

Reviewer #4: No

4. Have the authors made all data underlying the findings in their manuscript fully available?

Reviewer #3: Yes

Reviewer #4: No

5. Is the manuscript presented in an intelligible fashion and written in standard English?

Reviewer #3: Yes

Reviewer #4: No

6. Review Comments to the Author

Reviewer #3: Thank you for responding to my comments. I believe the authors dedicated a lot of time going through all the reviews. Congrats on that! I still believe the authors did not explain the randomization method. Mentioning that randomization was performed is not enough, how was it done? They are several randomization methods? Which one did you use?

Also, the authors compared pre and post in exp and control group (fine); compared pre (cont) vs pre (int) and post (cont) vs (pos int), which is also fine but the most important comparison was not performed. The mean (median) difference between group is the one that is really informative and should be compared to the MCID.

Reviewer #4: Abstract

Background

Research gap was not stated

method

major statistical analysis was not specified

Main body

major statistical analysis was not specified

results

The major statistical model output were not presented

7. PLOS authors have the option to publish the peer review history of their article (what does this mean?). If published, this will include your full peer review and any attached files.

Reviewer #3: No

Reviewer #4: No

---

## [Author Response · Author response to Decision Letter 2]

13 Sep 2021

Here is a point-by-point response to the reviewers’ comments and concerns.

No. Comments from Reviewer 3 

1. Thank you for responding to my comments. I believe the authors dedicated a lot of time going through all the reviews. Congrats on that! I still believe the authors did not explain the randomization method. Mentioning that randomization was performed is not enough, how was it done? They are several randomization methods? Which one did you use? 

Firstly, we want to apologize because we are not clear on the type of randomization intended by the reviewer. However, this is the best explanation that we can give for this remark.

A randomised controlled trial, pretest-posttest control group design was conducted among 24 traffic police riders who ride a high-powered motorcycle (Honda CBX 750). A simple random sampling was used in this study which the subjects were randomly assigned to the control group (12 riders) and experimental group (12 groups). Data collection commenced in March 2020 and finished in July 2020. 

Further explanation had been explained in line 138-165.

2. Also, the authors compared pre and post in exp and control group (fine); compared pre (cont) vs pre (int) and post (cont) vs (pos int), which is also fine but the most important comparison was not performed. The mean (median) difference between group is the one that is really informative and should be compared to the MCID. Thank you for pointing this out. However, this MCID actually will be discussed in details in our future paper. The main objective for this work is to evaluate the effect of lumbar support with a built-in massager system on spinal angle profiles among traffic police riders which focus on comparison of pretest-posttesst and between control-experimental groups. Nevertheless, for future studies, this will be a great insight and we will be sure to use it.

No. Comments from Reviewer 4 

1. Reviewer #4: Abstract

Background

Research gap was not stated

method

major statistical analysis was not specified

Main body

major statistical analysis was not specified

results

The major statistical model output were not presented 

We really appreciate your valuable comments here. We find it really useful to improve our manuscript. After discussion with our co-authors, we have improved our manuscript (in the 2nd revision) based on the given comments as below: -

i. The context and research gap of traffic police riders suffered low back pain, and MSD which no research has been done on the spinal riding posture during a motorcycle ride was explained in line 72-87.

ii. The context and research gap in which there is a lack of data on the effectiveness of this intervention seat, lumbar support with a built-in massager system, in an in-field setting (on-the-road) was explained in line 96-99.

The major statistical analysis used in this study was the Mann-Whitney test and Wilcoxon signed-ranked test. The details of the statistical analysis were explained in line 233-243.

The sampling technique was explained in Participant Recruitment (line 132-158).

However, the similar comments are asked again in this 3rd revision. Thus, we decided to explained this in the abstract since it has a word abstract in the beginning of the comments.

Traffic police riders are exposed to prolonged static postures causing significant angular deviation of the musculoskeletal, including the lumbar angle (L1-L5). This postural alteration contributes to awkward posture, musculoskeletal disorders and spinal injury, especially in the lower back area, as it is one of the most severe modern diseases nowadays. Thus, the study aimed to evaluate the effect of lumbar support with a built-in massager system on spinal angle profiles among traffic police riders. A randomised controlled trial (pre-testpost-test control design) was used to assess spinal angle pattern while riding the high-powered motorcycle for 20 minutes. Twenty-four traffic police riders were randomly selected to participate and 12 riders were assigned to the control group and 12 riders to the experimental group. The pre-test and post-test were conducted at a one-week interval. Each participant was required to wear a TruPosture Smart Shirt (to monitor spinal posture). The TruPosture Apps recorded the spinal angle pattern. The data indicated that the police riders using motorcycle seat with lumbar support and built-in massager system showed a huge improvement in maintaining posture which only involves slight spinal angle deviation changes from the spinal reference angle throughout the 20 minutes ride. The data collected then were analysed using the Mann-Whitney test and Wilcoxon signed-ranked test to verify a statistically significant difference between and within the control and experimental groups. There were significant differences in all sensors between the control group and experimental groups (p<0.05) and within the experimental group. According to the findings, it can be said that the ergonomic intervention prototype (lumbar support with built-in massager system) successfully helps to maintain and improve the natural curve of the spinal posture. This indirectly would reduce the risk of developing musculoskeletal disorders and spinal injury among traffic police riders.

We do provide the justification as requested as based on our understanding. Please do advise us in details if we need further revision as we feel your valuable comments will improve further our manuscript. Thank you.

---

## [Decision Letter · Decision Letter 3]

6 Oct 2021

Effectiveness of lumbar support with built-in massager system on spinal angle profiles among high-powered traffic police motorcycle riders: A randomised controlled trial

PONE-D-21-13337R3

Dear Dr. Karuppiah,

We’re pleased to inform you that your manuscript has been judged scientifically suitable for publication and will be formally accepted for publication once it meets all outstanding technical requirements.

Kind regards,

Ahmed Mancy Mosa, Ph.D.

Academic Editor

PLOS ONE

Additional Editor Comments (optional):

Reviewers' comments:

Reviewer's Responses to Questions

**Comments to the Author**

1. If the authors have adequately addressed your comments raised in a previous round of review and you feel that this manuscript is now acceptable for publication, you may indicate that here to bypass the “Comments to the Author” section, enter your conflict of interest statement in the “Confidential to Editor” section, and submit your "Accept" recommendation.

Reviewer #3: (No Response)

Reviewer #4: All comments have been addressed

2. Is the manuscript technically sound, and do the data support the conclusions?

Reviewer #3: (No Response)

Reviewer #4: Yes

3. Has the statistical analysis been performed appropriately and rigorously? 

Reviewer #3: (No Response)

Reviewer #4: Yes

4. Have the authors made all data underlying the findings in their manuscript fully available?

Reviewer #3: (No Response)

Reviewer #4: Yes

5. Is the manuscript presented in an intelligible fashion and written in standard English?

Reviewer #3: (No Response)

Reviewer #4: Yes

6. Review Comments to the Author

Reviewer #3: (No Response)

Reviewer #4: Remove" The dependent variable of this study was the spinal angle (0th and 20th

minutes). Indicate how outcome of interest is assessed before data analysis?

7. PLOS authors have the option to publish the peer review history of their article (what does this mean?). If published, this will include your full peer review and any attached files.

Reviewer #3: No

Reviewer #4: No

---

## [Editor Report · Acceptance letter]

11 Oct 2021

PONE-D-21-13337R3 

Effectiveness of lumbar support with built-in massager system on spinal angle profiles among high-powered traffic police motorcycle riders: A randomised controlled trial 

Dear Dr. Karupiah:

I'm pleased to inform you that your manuscript has been deemed suitable for publication in PLOS ONE. Congratulations! Your manuscript is now with our production department. 

Kind regards, 

on behalf of

Dr. Ahmed Mancy Mosa 

Academic Editor

PLOS ONE